# Efficacy and Safety of Different Courses of Tongxinluo Capsule as Adjuvant Therapy for Coronary Heart Disease after Percutaneous Coronary Intervention: A Systematic Review and Meta-Analysis of Randomized Controlled Trials

**DOI:** 10.3390/jcm11112991

**Published:** 2022-05-25

**Authors:** Jiaqi Hui, Rong Yuan, Pengqi Li, Qiqi Xin, Yu Miao, Xiaoxu Shen, Fengqin Xu, Weihong Cong

**Affiliations:** 1Laboratory of Cardiovascular Diseases, Xiyuan Hospital, China Academy of Chinese Medical Sciences, Beijing 100091, China; huijiaqi77@163.com (J.H.); yuanrong427@163.com (R.Y.); lipengqi37@126.com (P.L.); xinqiqiyuan@126.com (Q.X.); miaoyu0609@sina.com (Y.M.); 2National Clinical Research Center for Chinese Medicine Cardiology, Beijing 100091, China; 3Institute of Geriatrics, Xiyuan Hospital, China Academy of Chinese Medical Sciences, Beijing 100091, China; 4Dongzhimen Hospital, Beijing University of Chinese Medicine, Beijing 100700, China; ghxiaoxushen@sina.com

**Keywords:** Tongxinluo capsule, coronary heart disease, percutaneous coronary intervention, meta-analysis, systematic review, traditional Chinese medicine

## Abstract

Tongxinluo capsule (TXLC) is a widely used traditional Chinese medicine for coronary heart disease (CHD). However, the efficacy and safety of different courses of TXLC for CHD after percutaneous coronary intervention (PCI) have not been systematically evaluated yet. The Cochrane Library, PubMed, Embase, China National Knowledge Infrastructure, Wanfang Database, and Chinese Scientific Journal Database were searched from the inception to 26 August 2021. A meta-analysis was performed using a fixed- or random-effects model. The risk of adverse cardiovascular events, mortality, or adverse effects was evaluated by risk ratio (RR) with 95% confidence interval (CI). Thirty-four studies involving 3652 patients were finally included. After the 6-month treatment, compared with conventional treatment alone, TXLC combined with conventional treatment achieved better efficacy in lowering the risk of angiographic restenosis (RR = 0.37, 95% CI = 0.28–0.48, *p* < 0.001), myocardial infarction (RR = 0.38, 95% CI = 0.25–0.60, *p* < 0.001), heart failure (RR = 0.32, 95% CI = 0.18–0.56, *p* < 0.001), angina (RR = 0.26, 95% CI = 0.17–0.38, *p* < 0.001), revascularization (RR = 0.20, 95% CI = 0.09–0.46, *p* < 0.001), all-cause mortality (RR = 0.24, 95% CI = 0.10–0.58, *p* = 0.001), and mortality due to any cardiovascular event (RR = 0.27, 95% CI = 0.09–0.80, *p* = 0.018). After the 12-month treatment, TXLC reduced the recurrence risk of angina (RR = 0.40, 95% CI = 0.20–0.80, *p* = 0.009). However, there was no difference in any outcomes after the 3-month treatment. Besides, no difference was found in the incidence of adverse effects after the 3-month and 6-month treatments (3 months: RR = 0.73, 95% CI = 0.35–1.56, *p* = 0.418; 6 months: RR = 1.71, 95% CI = 0.74–3.93, *p* = 0.209). The certainty of evidence ranged from very low to moderate due to the risk of bias, inconsistency, and imprecision. TXLC showed beneficial effects on reducing the adverse cardiovascular events without compromising safety for CHD patients after PCI on the 6-month course. However, due to the unavoidable risk of bias, more high-quality and long-term studies are still needed to further evaluate the efficacy and safety of TXLC in many countries, not only in China.

## 1. Introduction

Cardiovascular disease (CVD) is the leading cause of death worldwide [1]. The latest data from the Global Burden of Diseases (GBD) 2019 showed that the burden of CVD had been rising globally for more than three decades [2]. Coronary heart disease (CHD) is the most common CVD with high mortality and morbidity worldwide. CHD-associated mortality is expected to increase by 100% among men and by 80% among women from 1990 to 2020 [3,4]. In recent years, percutaneous coronary intervention (PCI) as a coronary revascularization procedure has been widely selected as one of the treatment strategies for CHD patients. Open occluded coronary arteries through cardiac catheterization could improve myocardial perfusion and relieve clinical symptoms. According to the Guidelines for Rational Drug Use of Coronary Heart Disease (second edition), the conventional treatments for CHD after PCI mainly include antiplatelet, anticoagulation, heart rate control, lipid-regulating, and anti-ischemic therapies [5]. However, many complications still occur after PCI, such as stent thrombosis and in-stent restenosis, thus increasing the incidence of adverse cardiovascular events, such as no-reflow and slow flow, ischemia–reperfusion injury, perioperative myocardial injury, target lesion revascularization, and angina [6,7,8], all of which influence the long-term prognosis and rehabilitation of patients.

CHD belongs to the category of “chest pain” and “palpitation” in traditional Chinese medicine (TCM). TCM has a history of thousands of years and has provided a rich basis for modern drug discovery and development. Tongxinluo capsule (TXLC), a pharmaceutical product developed according to the TCM theory and registered with the China Food and Drug Administration (CFDA) in 1996 [9], have been widely used for CHD in China and proved to have beneficial effects both in clinical and experimental studies. It consists of *Panax ginseng* C.A.Mey. (Ren Shen), *Paeonia lactiflora* Pall. (Chi Shao), *Ziziphus jujuba* Mill. (Suan Zao Ren), *Santalum album* L. (Tan Xiang), *Dalbergia odorifera* T.C.Chen (Jiang Xiang), *Dryobalanops aromatica* C.F.Gaertn. (Bing Pian), *Scolopendra subspinipes murilans* L. Koch (Wu Gong), *Hirudo nipponica* Whitman (Shui Zhi), *Eupolyphaga sinensis* Walker (Tu Bie Chong), *Cryptotympana pustulata* Fabricius (Chan Tui), *Buthus martensii* Karsch (Quan Xie), and *Boswellia carteri* Birdw. (Ru Xiang), with definite effects of benefiting the Qi and activating blood circulation, dredging collaterals, and relieving pain. The mechanism might be related to increasing cardiac output, reducing myocardial oxygen consumption, protecting endothelial cells, preventing apoptosis, enhancing angiogenesis, regulating blood lipids, inhibiting vascular inflammation, reducing myocardial remodeling, etc. [9,10,11]. In high-performance liquid chromatography (HPLC) analysis, the similarity of the fingerprints of each batch of TXLC is above 95%, indicating the stable and controllable quality [12,13]. Evidence from clinical trials indicated that TXLC might reduce the risks of restenosis, target lesion revascularization, angina attacks, and other cardiac events [13,14,15]. Experimental evidence revealed that TXLC could prevent atherosclerosis development, increase the stability of plaques, protect endothelial structure and function, inhibit vascular inflammation, etc. [11,16,17].

To our knowledge, in-stent restenosis is defined as a 50% reduction in the luminal diameter after PCI. Regardless of whether bare metal stents or drug-eluting stents were used for in-stent restenosis lesions, 47% of the lesions had neoatherosclerosis, which drove the target lesion revascularization rates [18]. One study [7] reported that in-stent restenosis was detected in 16% of the acute coronary syndrome (ACS) patients after PCI at 6–24 months. A systematic review and meta-analysis [14] reported that TXLC significantly reduced the risks of angiographic restenosis, myocardial infarction, heart failure, angina, and revascularization. However, the treatment course of TXLC in the included studies was diverse. Up till now, few systematic reviews have been conducted on the efficacy and safety of TXLC specific to different treatment courses for CHD after PCI. In the present study, published reports of clinical trials about TXLC as the adjuvant therapy were collected comprehensively and systematically, and the efficacy and safety of TXLC in the treatment of CHD after PCI were re-evaluated as well to provide a more accurate and reliable reference for the clinic.

## 2. Materials and Methods

This review was designed, conducted, and reported based on the guidelines of the Preferred Reporting Items for Systematic Reviews and Meta-Analyses (PRISMA) statement [19]. The protocol of this systematic review and meta-analysis was registered and accepted for inclusion in the International Prospective Register of Systematic Reviews (PROSPERO, CRD42021232519). Since this study is a systematic review and meta-analysis of human intervention studies, it did not require approval from an ethics committee.

### 2.1. Search Strategy

A systematic search of the Cochrane Library, PubMed, Embase, China National Knowledge Infrastructure (CNKI), Wanfang Database, and Chinese Scientific Journal Database (Chinese VIP Information) was conducted to identify relevant studies from their inception to 26 August 2021 with no language or ethnicity restrictions. The medical subject heading and the following free words were searched: “Tong-xin-luo,” “TXL,” “Tongxinluo,” “Tong Xin Luo,” “percutaneous coronary intervention,” “PCI,” etc. The retrieval expressions were formed by logical connection using AND or OR. The full search strategy was shown in Appendix A.

### 2.2. Type of Studies

The inclusion criteria were as follows:Patients: patients with a confirmed diagnosis of CHD should have received a successful PCI;Intervention: patients in the treatment group were given TXLC combined with conventional treatment with the treatment duration of no less than three months, and TXLC should have been used after PCI;Control: patients in the control group were given conventional treatment alone (including aspirin, clopidogrel, atorvastatin, etc., according to clinical signs and symptoms) or conventional treatment plus placebo;Outcomes: the outcomes of these studies must include at least one of the following indicators: angiographic restenosis, myocardial infarction, heart failure, angina, all-cause mortality, mortality due to any cardiovascular event, revascularization, and adverse effects;Study type: randomized controlled trials (RCTs).

The exclusion criteria were as follows: Observational studies, editorials, commentaries, review articles, case reports, animal experiments, and single-arm trials;Incomplete or serious errors in data;Patients in the treatment group were given TXLC before PCI;Intervention measures included other Chinese patent medicines, TCM injections, acupuncture, ear points, Qi gong, Tai Chi, cupping, or other TCM therapy;No relevant outcomes were reported.

### 2.3. Type of Participants

The patients met the abovementioned inclusion criteria with no age, gender, or race restrictions. The reported CHD in RCTs included ACS, acute myocardial infarction (AMI), ST-segment elevation myocardial infarction (STEMI), and unstable angina (UA).

### 2.4. Type of Interventions

The daily dose of TXLC depends on the clinical signs and symptoms of patients. In general, the dosage ranged from 0.78 g to 1.14 g. Conventional therapy must be consistent both in the treatment group and the control group. Both TXLC and conventional treatment were used in the TXLC group, and conventional treatment alone was used in the control group for comparison. Other intervention measures were excluded except for TXLC and conventional treatment in the TXLC group.

### 2.5. Type of Outcome Measures

The primary outcomes included the occurrence of angiographic restenosis, myocardial infarction, heart failure, angina, and all-cause mortality. The secondary outcomes included mortality due to any cardiovascular event, the occurrence of revascularization (PCI or coronary artery bypass graft surgery (CABG)), and adverse effects (epigastric or stomach discomfort, nausea, vomiting, constipation, and bleeding events).

### 2.6. Study Selection and Data Extraction

Two review authors (J.H. and R.Y.) independently conducted the literature search, study selection, and data extraction. The search results were imported to EndNote X9 (Clarivate Analytics, Philadelphia, PA, USA), and duplicates were removed. Titles and abstracts were screened for initial study inclusion, and a full-text review was conducted to ensure the studies met the inclusion criteria. Any disagreements were resolved through discussion or consultation with a third reviewer. If there was any unclear or missing information, the original authors were contacted for additional information or clarification where required. If the literature had multiple endpoint indicators, the longest one was adopted. The data of the study characteristics were extracted, including the author, year, age, gender, sample size, participants, stent type, intervention, treatment courses, outcomes, study design, and so forth.

### 2.7. Methodological Quality Assessment

The risk of bias of the included studies was assessed according to the guidance in the Cochrane Collaboration’s Risk of Bias Handbook for RCTs. The risk of bias assessment was presented for the following domains: sequence generation, allocation concealment, blinding, incomplete outcome data, selective outcome reporting, and other bias. The risk of bias for each domain can be categorized as follows: low risk of bias, high risk of bias, and unclear risk of bias. To confirm and validate the methods of allocation concealment and the randomization procedure, the original authors were contacted. If the original authors did not get in touch, any disagreements were resolved through discussion. In addition, we assessed the quality of included evidence using the Grading of Recommendations, Assessment, Development and Evaluations (GRADE)pro Guideline Development Tool (GDT) online software, including the total number of studies, study design, risk of bias, inconsistency, indirectness, imprecision, other considerations, summary of findings, and importance. The evidence was classified as high, moderate, low, or very low.

### 2.8. Data Synthesis and Analysis

Dichotomous data including the risk of angiographic restenosis, myocardial infarction, heart failure, angina, mortality, revascularization, and adverse effects after PCI were determined by using risk ratio (RR) with 95% confidence interval (CI). Heterogeneity across studies was tested with the I^2^ statistic, a quantitative measure of inconsistency across studies. Studies with an I^2^ statistic of 25~50% were considered to reflect low heterogeneity, those with an I^2^ statistic of 50~75% were considered to reflect moderate heterogeneity, and those with an I^2^ statistic of >75% were considered to reflect high heterogeneity [20]. A fixed-effects model was used if there was no substantial heterogeneity between the studies (I^2^ < 50% or *p* > 0.1), otherwise a random-effects model was used. These results were visualized with forest plots. Two-sided *p* < 0.05 was considered statistically significant. A stratified meta-analysis was conducted to evaluate the effect of different treatment courses of TXLC: 3 months, 6 months, and 12 months. Subgroup and sensitivity analysis were conducted to explore the sources of significant heterogeneity and robustness of the results. The risk of publication bias was assessed through funnel plots, Egger’s test, and Begg’s test when the number of RCTs was ≥ 10. All analyses were carried out by using the STATA statistical software (version 12.0; Stata Corp LP, College Station, TX, United States), complemented by Review Manager 5.3 (The Nordic Cochrane Centre, The Cochrane Collaboration, Copenhagen, Denmark). If *p* > 0.05, there is no publication bias and vice versa.

## 3. Results

### 3.1. Study Selection

The systematic search identified a total of 4883 articles. After removing duplicates, 4165 articles remained for first-stage screening. After first-stage screening by reviewing the titles and abstracts, 3950 articles were excluded and 215 articles were identified to assess the full text for eligibility. After the second stage of screening, 181 articles were excluded. Finally, 34 studies [21,22,23,24,25,26,27,28,29,30,31,32,33,34,35,36,37,38,39,40,41,42,43,44,45,46,47,48,49,50,51,52,53,54] met all the inclusion criteria and were included in the systematic review. Three studies [40,43,51] were not included in the meta-analysis since the outcome indicators were assessed after follow-up instead of after treatment, and there were inconsistencies between treatment duration and follow-up time. Therefore, 34 articles were included in the systematic review while only 31 articles were finally included in the meta-analysis. The details of the study selection process are shown in Figure 1.

### 3.2. Study Characteristics

The main characteristics of the included 34 articles are summarized in Table 1. In total, 3652 patients were included, of whom 2072 were males and 1236 were females, and two studies [33,35] did not clarify the number of males or females. The age of the patients ranged from 18 to 85 years. Five trials [23,27,44,47,48] had a treatment course of 3 months, 24 trials [21,22,24,25,28,29,30,31,32,33,34,35,36,37,38,39,41,42,45,46,49,52,53,54] had a treatment course of 6 months, and three trials [26,36,50] had a treatment course of 12 months. One study [36] reported the efficacy and safety of TXLC after the 6- and 12-month treatment. Three studies [40,43,51] reported the inconsistencies between treatment duration and follow-up time. For stent type, only two studies provided specific data: one study [46] used bare metal stents and the other [50] used drug-eluting stents. For outcome measures, the included 34 studies fully or partially reported the primary outcomes and secondary outcomes. Adverse effects were reported in nine trials, of which one [44] referred to rash, nausea or vomiting, abdominal distension, and constipation, one [33] reported drug allergy and gingival bleeding, two [30,39] reported no abnormal liver and kidney function, four [22,30,33,39] reported epigastric discomfort, two [23,48] reported stomach discomfort, one [22] reported dizziness, and two [27,31] reported no specific symptoms. The general characteristics of the 34 studies were summarized in Table 1.

### 3.3. Risk of Bias

The risk of bias of the included 34 trials is shown in Figure 2 and Figure 3. Sixteen trials (47%) presented an unclear risk of bias, and two trials (6%) presented a high risk of selection bias in random sequence generation. Besides, two trials (6%) presented a high risk of bias, and 31 (91%) presented an unclear risk of selection bias in allocation concealment. The vast majority of involved trials, except for one study [30], included complete outcome data and presented a low risk of attrition bias. As for the blinding of participants and personnel or outcome assessment, the unclear risk of performance and detection bias existed in all the trials. In addition, seven trials (21%) presented a high risk of selective reporting bias. In brief, the quality of the included RCTs was relatively not high.

### 3.4. Primary Outcomes

#### 3.4.1. Occurrence of Angiographic Restenosis

Nineteen trials reported the occurrence of angiographic restenosis after PCI in Figure 4. Seventeen trials consisting of 1772 participants showed that the incidence of angiographic restenosis was significantly reduced in the TXLC group compared with the control group after the 6-month treatment (RR = 0.37, 95% CI = 0.28–0.48, *p* < 0.001) with no evidence of heterogeneity (I^2^ = 0.0%), while there was no difference between two groups after the 3-month treatment (RR = 0.20, 95% CI = 0.01–4.04, *p* = 0.294). Two trials [36,50] including 294 participants demonstrated that the incidence of angiographic restenosis was significantly reduced in the TXLC group compared with the control group after the 12-month treatment (RR = 0.52, 95% CI = 0.34–0.80, *p* = 0.003), but represented statistical heterogeneity (I^2^ = 52.8%). For this result, a random effects model was used to re-evaluate the efficacy of TXLC on the 12-month course, and the difference became not statistically significant (RR = 0.44, 95% CI = 0.16–1.22, *p* = 0.113, I^2^ = 52.8%).

#### 3.4.2. Occurrence of Myocardial Infarction

Fifteen trials reported the occurrence of myocardial infarction after PCI in Figure 5. Ten trials involving 1068 participants reported that the incidence of myocardial infarction was significantly reduced in the TXLC group compared with the control group after the 6-month treatment (RR = 0.38, 95% CI = 0.25–0.60, *p* < 0.001). Five studies including 519 participants reported that the incidence of myocardial infarction was lower in the TXLC group than in the control group after the 3- or 12-month treatment, but the difference between two groups was not statistically significant (3 months: RR = 0.33, 95% CI = 0.04–3.18, *p* = 0.340; 12 months: RR = 0.44, 95% CI = 0.16–1.25, *p* = 0.122). There was no evidence of heterogeneity (I^2^ = 0.0%).

#### 3.4.3. Occurrence of Heart Failure

Six trials provided data on the occurrence of heart failure after PCI in Figure 6. Four studies including 482 participants demonstrated that the incidence of heart failure was significantly reduced in the TXLC group compared with the control group after the 6-month treatment (RR = 0.32, 95% CI = 0.18–0.56, *p* < 0.001) with no evidence of heterogeneity (I^2^ = 0.0%), while there was no difference between two groups after the 3- or 12-month treatment (3 months: RR = 0.24, 95% CI = 0.06–1.04, *p* = 0.057; 12 months: RR = 0.50, 95% CI = 0.05–5.36, *p* = 0.576).

#### 3.4.4. Occurrence of Angina

A meta-analysis of 14 trials was performed to estimate the risk of occurrence of angina after PCI in Figure 7. Eleven studies including 1236 participants demonstrated that the incidence of angina was significantly reduced in the TXLC group compared with the control group after the 6-month treatment (RR = 0.26, 95% CI = 0.17–0.38, *p* < 0.001) with no evidence of heterogeneity (I^2^ = 0.0%). Two trials [36,50] including 294 participants demonstrated that the incidence of angina was significantly reduced in the TXLC group compared with the control group after the 12-month treatment (RR = 0.40, 95% CI = 0.20–0.80, *p* = 0.009) with no evidence of heterogeneity (I^2^ = 0.0%). One study [44] involving 103 participants reported that the incidence of angina was lower in the TXLC group than in the control group after the 3-month treatment, but the difference between the treatment group and the control group was not statistically significant (RR = 0.25, 95% CI = 0.03–2.20, *p* = 0.214).

#### 3.4.5. All-Cause Mortality

Twelve studies reported the all-cause mortality after PCI in Figure 8. Nine trials including 1100 participants showed that the all-cause mortality was significantly reduced in the TXLC group compared with the control group after the 6-month treatment (RR = 0.24, 95% CI = 0.10–0.58, *p* = 0.001) with no evidence of heterogeneity (I^2^ = 0.0%). Three studies [26,36,47] including 302 participants reported that the all-cause mortality was lower in the TXLC group than in the control group after the 3- or 12-month treatment, but the difference between the treatment group and the control group was not statistically significant (3 months: RR = 0.32, 95% CI = 0.01–7.61, *p* = 0.483; 12 months: RR = 0.51, 95% CI = 0.07–3.67, *p* = 0.504, I^2^ = 0.0%).

### 3.5. Secondary Outcomes

#### 3.5.1. Mortality Due to Any Cardiovascular Event

Nine studies reported the risk of mortality due to any cardiovascular event in Figure 9. Seven trials including 862 participants showed that the mortality due to any cardiovascular event was significantly reduced in the TXLC group compared with the control group after the 6-month treatment (RR = 0.27, 95% CI = 0.09–0.80, *p* = 0.018) with no evidence of heterogeneity (I^2^ = 0.0%). Two trials [26,47] showed the slight effectiveness of TXLC treatment on mortality due to any cardiovascular event, which yielded RRs with the large CIs indicating no statistical significance after the 3- or 12-month treatment (3 months: 0/30 vs. 1/29, RR = 0.32, 95% CI = 0.01–7.61, *p* = 0.483; 12 months: 1/35 vs. 2/28; RR = 0.40, 95% CI = 0.04–4.19, *p* = 0.444).

#### 3.5.2. Revascularization

Eight trials showed the incidence of revascularization after PCI in Figure 10. Five trials including 608 participants demonstrated that TXLC treatment was effective in reducing the risk of revascularization (RR = 0.20, 95% CI = 0.09–0.46, *p* < 0.001) after the 6-month treatment with no evidence of heterogeneity (I^2^ = 0.0%), while there was no difference between the two groups after the 3- or 12-month treatment (3 months: RR = 0.60, 95% CI = 0.08–4.47, *p* = 0.621, I^2^ = 0.0%; 12 months: RR = 0.12, 95% CI = 0.01–2.14, *p* = 0.147, I^2^ = 0.0%).

#### 3.5.3. Adverse Effects

Nine studies including 1054 participants reported adverse effects after the TXLC treatment in Figure 11. The incidence of adverse effects was lower in the TXLC group (10.5/179.5) than in the control group (14.5/180.5) after the 3-month treatment, while the incidence of adverse effects was slightly higher in the TXLC group (14/350) after the 6-month treatment. However, the difference between the treatment group and the control group was not statistically significant (3 months: RR = 0.73, 95% CI = 0.35–1.56, *p* = 0.418, I^2^ = 0.0%; 6 months: RR = 1.71, 95% CI = 0.74–3.93, *p* = 0.209, I^2^ = 0.0%). No study reported adverse effects on the 12-month course.

### 3.6. Publication Bias

Publication bias was observed with inspection of the funnel plot, Egger’s test, and Begg’s test in Figure 12. The asymmetrical funnel plots of angiographic restenosis and myocardial infarction on the 6-month course suggested that a high risk of publication bias might exist (angiographic restenosis: Egger’s test: *p* = 0.001, after the trim-and-fill computation: *p* = 0.831; myocardial infarction: Egger’s test: *p* = 0.016, after the trim-and-fill computation: *p* = 0.999), and the symmetrical funnel plot of angina on the 6-month course indicated no risk of publication bias (Egger’s test: *p* = 0.165).

### 3.7. Subgroup and Sensitivity Analysis

Although statistical heterogeneity was found in the incidence of angiographic restenosis after the 12-month treatment (I^2^ = 52.8%), it was not possible to conduct subgroup and sensitivity analysis because of insufficient numbers and the low quality of included trials. Other outcome measures yielded the results with no evidence of heterogeneity (I^2^ = 0.0%), thus there was no need to perform subgroup analysis. Sensitivity analyses were conducted to explore the robustness of the results including angiographic restenosis, myocardial infarction, heart failure, revascularization, angina, all-cause mortality, and mortality due to any cardiovascular event. The pooled effect estimates showed no significant difference for mortality due to any cardiovascular event after excluding the study of Lu, 2017 [37], which indicated that the result was not robust. After rechecking the studies that reported mortality due to any cardiovascular event after the 6-month treatment, only Lu’s study did not use randomization methods for grouping. The participants in Lu’s study were randomized to the treatment group and the control group according to receiving TXLC or not. The high risk of selection bias might exaggerate the effect of TXLC in lowering the risk of mortality due to any cardiovascular event. In addition, individual study exclusion did not substantially modify the pooled effect estimates of other outcomes.

### 3.8. Quality of the Evidence

All outcomes on the 3-, 6-, and 12-month courses were assessed using GRADEpro GDT, and the evidence profile is shown in Table 2. There was moderate-quality evidence on heart failure (6 months), angina (6 and 12 months), all-cause mortality (6 months), mortality due to any cardiovascular event (6 months), and revascularization (6 months); low-quality evidence on angiographic restenosis (6 and 12 months), myocardial infarction (3, 6, and 12 months), all-cause mortality (12 months), revascularization (3 months), and adverse effects (3 and 6 months); and very low-quality evidence on angiographic restenosis (3 months), heart failure (3 and 12 months), angina (3 months), all-cause mortality (3 months), mortality due to any cardiovascular event (3 and 12 months), and revascularization (12 months). The certainty of the evidence was not high, which might be attributed to the risk of bias, inconsistency, and imprecision, indicating that these estimates were uncertain and future studies might influence our confidence in the results.

## 4. Discussion

This meta-analysis provided evidence that TXLC as an adjuvant therapy had better clinical efficacy in the treatment of CHD after PCI compared with conventional treatment alone and could effectively reduce the risk of angiographic restenosis, myocardial infarction, heart failure, revascularization, angina, and lower all-cause mortality and mortality due to any cardiovascular event on the 6-month course. In the present study, the efficacy of TXLC specific to different treatment courses was evaluated, which is the novelty of this study. In the previous meta-analysis [14], the efficacy of TXLC in patients with CHD after PCI was proven regardless of the treatment course, but there was no difference between TXLC and the conventional treatment in reducing all-cause mortality and mortality due to cardiovascular events. In our study, after the 6-month treatment, there were significant differences between TXLC and conventional treatment in reducing the occurrence of cardiovascular events, all-cause mortality, and mortality due to cardiovascular events. After the 12-month treatment, TXLC better prevented the recurrence of angina, while not affecting other outcomes. Nevertheless, the efficacy of TXLC was not significant after the 3-month treatment. Therefore, TXLC showed obvious effects as an adjuvant therapy on the adverse cardiovascular events for patients with CHD after PCI on the 6-month course. For adverse effects, there is a debate about the safety of TXLC. Adverse effects such as epigastric or stomach discomfort, nausea and vomiting, constipation, bleeding events were observed in some RCTs [22,23,30,33,44,48], but not all [27,31,39]. It has not been reported whether the treatment course of TXLC is associated with the incidence of adverse reactions. In this meta-analysis, adverse effects were analyzed for different durations of TXLC, there is no statistical difference after the 3- or 6-month treatment, and no data after the 12-month treatment. More RCTs with a longer follow-up should be conducted to explore the optimal treatment course of TXLC in patients with CHD after PCI.

Race was an important factor in determining the outcome of CHD. Compared with White patients, Black race was an independent predictor of worse survival after CHD events [55,56,57]. A retrospective study [58] evaluated the 1-year major cardiovascular events in patients with CHD after PCI based on gender and race. The results showed that the major cardiovascular events were 1.3 to 1.5 times more likely to occur in Black men and women and in White women than in White men. Recently, mortality for CHD has experienced a longstanding decline compared to the previous period [59], which might be attributed to the widespread use of invasive strategies and medication [60]. There were extensive lesions and multiple “vulnerable” plaques in the coronary artery in patients with CHD, while PCI could only treat localized anatomic segments of obstructive atherosclerosis [61]. TCM exhibits a specific advantage in protecting the cardiovascular system through multi-targeting. Many clinical trials and experimental studies have shown the protective effects of TXLC in CHD. The mechanism might be related to inhibiting vascular inflammation, protecting endothelial cells, reducing myocardial remodeling, regulating blood lipids, preventing apoptosis, enhancing angiogenesis, retarding the progression of the mean intima media thickness, plaque area, and vascular remodeling of the carotid artery, and protecting human cardiomyocytes from ischemia/reperfusion injury [9,10,11]. A recent multicenter randomized double-blind parallel-group placebo-controlled study CAPITAL showed that TXLC retarded the progression of the mean intima media thickness and attenuated positive vascular remodeling of the carotid artery in addition to routine therapy [62]. TXLC could protect endothelial structure and function during oxygen–glucose–serum deprivation and restoration by activating the peroxisome proliferator-activated receptor-α/angiopoietin-like 4 pathway, which might be a strategy to defend endothelial barrier integrity against I/R injury in diabetic patients with AMI in the reperfusion era [16]. TXLC also focused on regulating ubiquitin to adjust the C-reactive protein levels and inflammation-related pathways to exert anti-inflammatory effects as well as improving hypoxia to protect myocardial cells in CHD patients [17]. In addition, TXLC could alleviate myocardial reperfusion injury by inhibiting directly cardiomyocyte apoptosis through the miR-128-3p/p70s6k1 pathway [63]. Additionally, TXLC could inhibit the hypoxia-induced endothelial-to-mesenchymal transition in human cardiac microvascular endothelial cells by activating the neuregulin-1/ErbB-PI3K/AKT signaling cascade and limiting excessive extracellular matrix deposition [64]. TXLC could also preserve the myocardium and promote cardiac function by promoting autophagy and inhibiting apoptosis through activating the AMPK signal pathway [65]. Therefore, these findings indicate that TXLC could promote heart repair and improve the prognosis of patients with CHD after PCI.

## 5. Limitations

This systematic review and meta-analysis have several limitations. First, the included trials had publication bias and not high methodologic quality: none of the included studies declared blinding; thirty-one studies did not describe allocation concealment, and two studies [29,37] did not use allocation concealment; sixteen studies did not describe a concrete random method, and two studies [29,37] did not use randomization methods for grouping, thus weakening the strength and trustworthiness of the clinical evidence of TXLC. Second, the treatment duration of most studies was relatively short, and the longest duration was 12 months; therefore, the effect of a longer duration of TXLC could not be evaluated in the present study. Third, due to the small number of trials included in the analysis on the 12-month course, the results should be interpreted with caution. Fourth, a considerable part of the included studies did not mention adverse reactions, which might affect safety evaluation. Fifth, the time of included studies ranged from 2006 to 2019; therefore, it was likely that there was a mixed use of bare metal stents and drug-eluting stents, which has quite a different performance in restenosis compared with drug-eluting stents. However, a stratified analysis was impossible to perform due to the insufficient data about the stent types in the included studies. Finally, the included participants in this review were all from China, and the universal applicability still could not be estimated in other races and regions.

Therefore, we suggest that future studies on the adjuvant treatment of CHD after PCI with TXLC should pay more attention to random sequence generation, allocation concealment, and implementation of blinding. More large-scale, prospective, randomized, double-blind, and long-term RCTs are needed to be conducted to improve the methodological flaws of the included studies. Besides, different stent types, treatment duration, dose–effect relationship, and racial differences should be considered as non-negligible factors affecting the efficacy and safety of TXLC. Future RCTs should pay attention to the incidence of adverse cardiovascular events, mortality, and adverse reactions in patients with long-term follow-up to provide better data to guide clinical decision-making.

## 6. Conclusions

In summary, compared with conventional treatment alone, TXLC plus conventional treatment exhibited a specific advantage in reducing the occurrence of adverse cardiovascular events without compromising safety on the 6-month course, and the effects of TXLC might be influenced by different time courses. In terms of the limitations of the current research, these results should be interpreted with caution. More multicenter, large-sample, high-quality RCTs with longer follow-up are required in the future, which will help to objectively evaluate the efficacy and safety of TXLC as an adjuvant treatment in patients with CHD after PCI and provide a more reliable reference for the clinic.

## Figures and Tables

**Figure 1 jcm-11-02991-f001:**
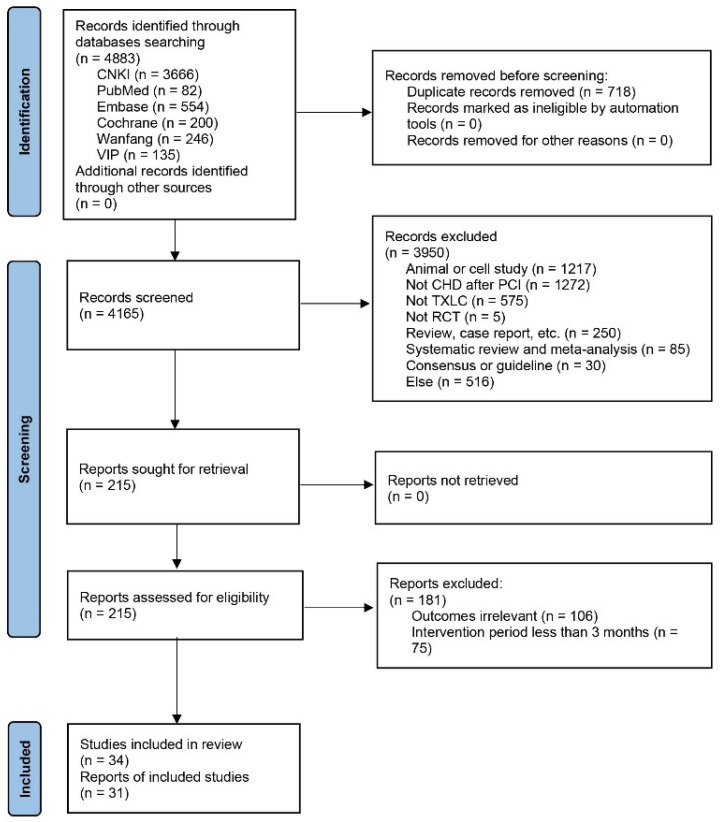
Flow diagram of the study selection process.

**Figure 2 jcm-11-02991-f002:**
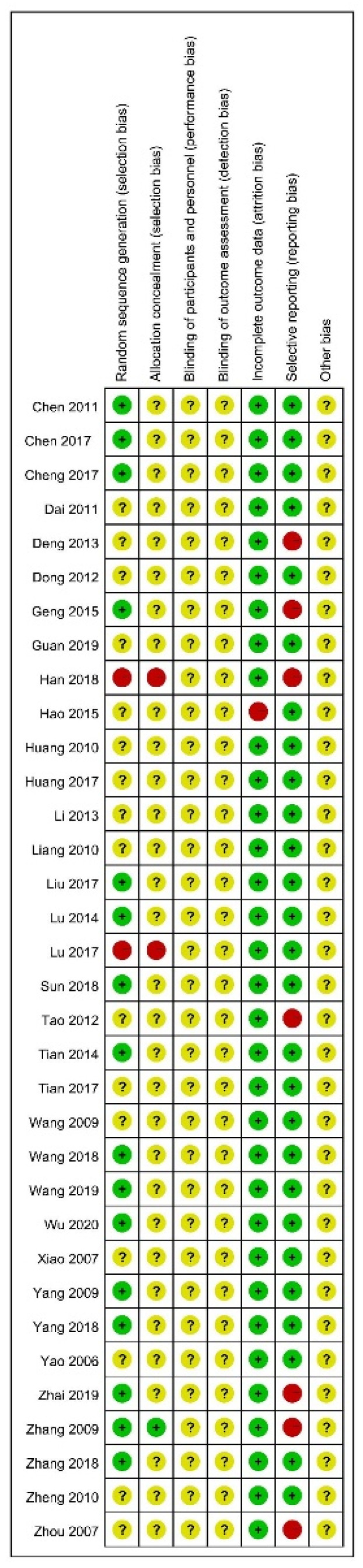
Risk of bias summary [21,22,23,24,25,26,27,28,29,30,31,32,33,34,35,36,37,38,39,40,41,42,43,44,45,46,47,48,49,50,51,52,53,54].

**Figure 3 jcm-11-02991-f003:**
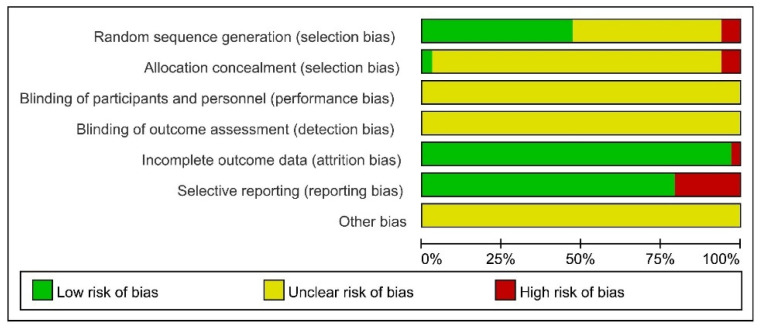
Risk of bias graph.

**Figure 4 jcm-11-02991-f004:**
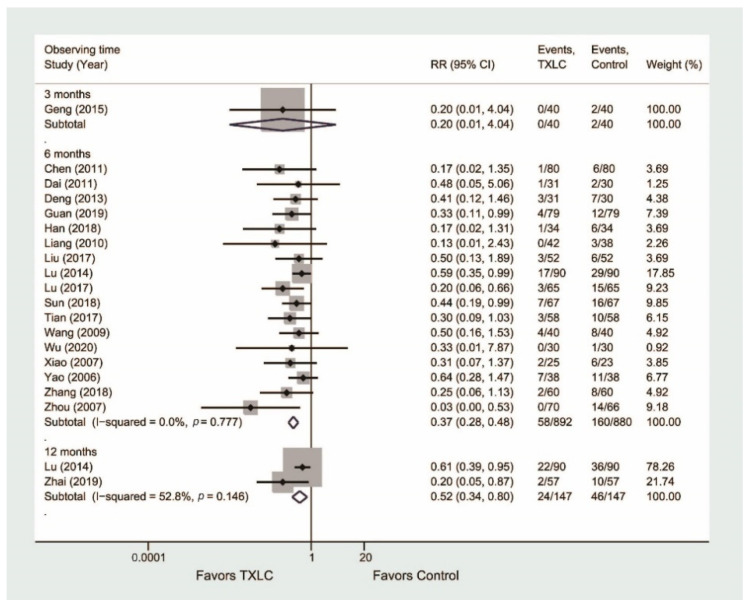
Forest plot describing the difference in occurrence of angiographic restenosis between the TXLC group and the control group on the 3- [27], 6- [21,24,25,28,29,34,35,36,37,38,41,42,45,46,49,52,54], and 12-month courses [36,50].

**Figure 5 jcm-11-02991-f005:**
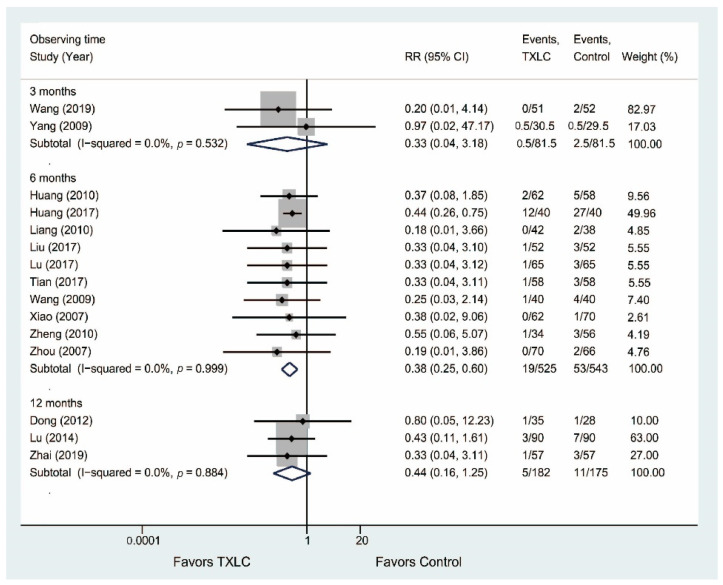
Forest plot describing the difference in occurrence of myocardial infarction between the TXLC group and the control group on the 3- [44,47], 6- [31,32,34,35,37,41,42,46,53,54], and 12-month courses [26,36,50].

**Figure 6 jcm-11-02991-f006:**
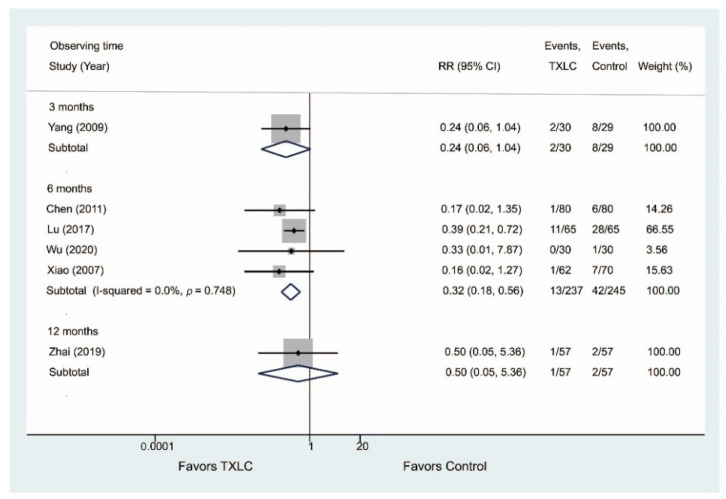
Forest plot describing the difference in occurrence of heart failure between the TXLC group and the control group on the 3- [47], 6- [21,37,45,46], and 12-month courses [50].

**Figure 7 jcm-11-02991-f007:**
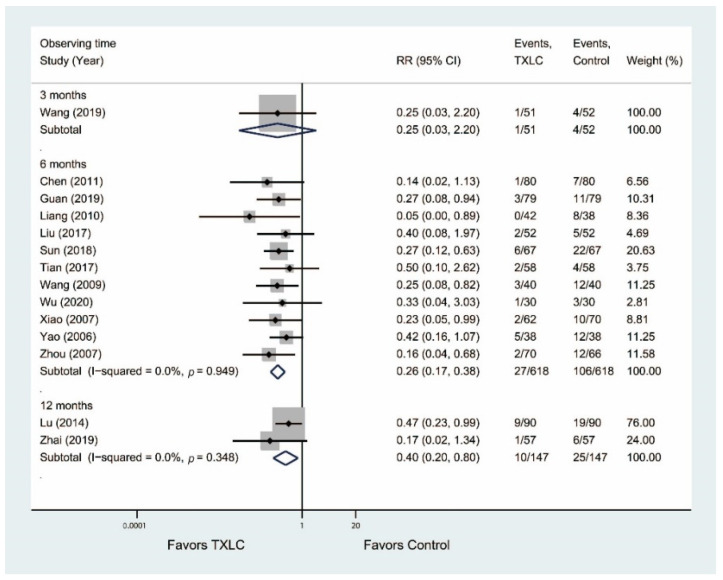
Forest plot describing the difference in occurrence of angina between the TXLC group and the control group on the 3- [44], 6- [21,28,34,35,38,41,42,45,46,49,54], and 12-month courses [36,50].

**Figure 8 jcm-11-02991-f008:**
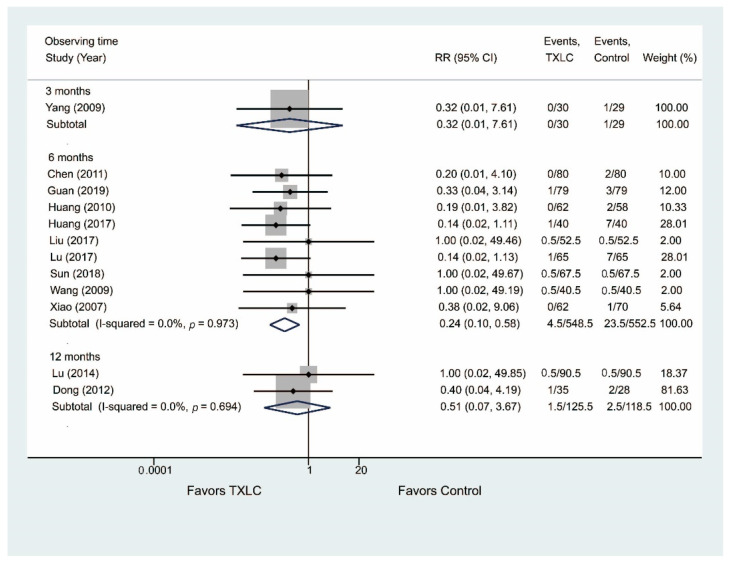
Forest plot describing the difference in all-cause mortality between the TXLC group and the control group on the 3- [47], 6- [21,28,31,32,35,37,38,42,46], and 12-month courses [26,36].

**Figure 9 jcm-11-02991-f009:**
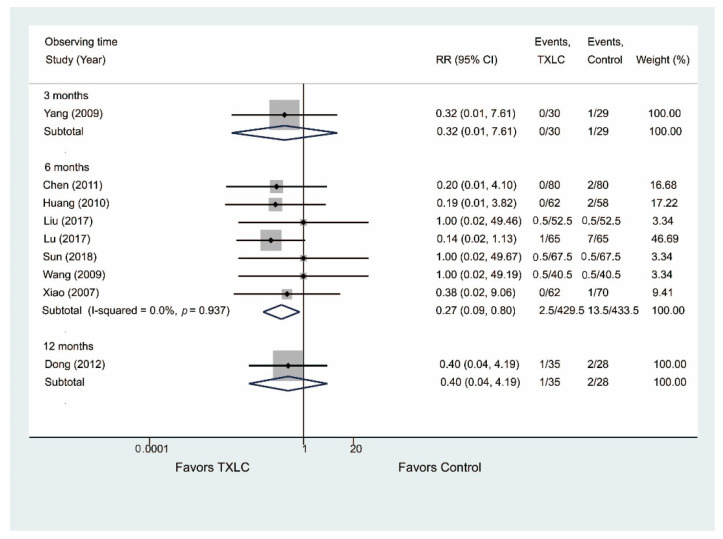
Forest plot describing the difference in mortality due to any cardiovascular event between the TXLC group and the control group on the 3- [47], 6- [21,31,35,37,38,42,46], and 12-month courses [26].

**Figure 10 jcm-11-02991-f010:**
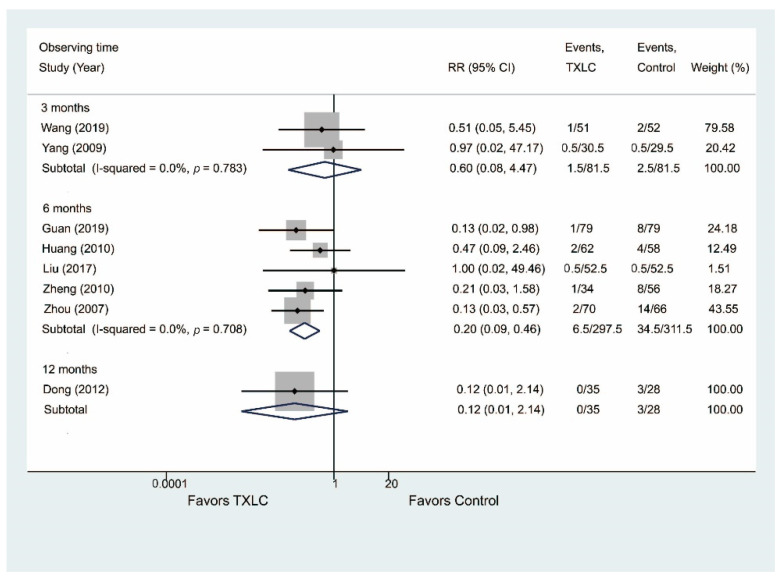
Forest plot describing the difference in revascularization, including PCI and CABG, between the TXLC group and the control group on the 3- [44,47], 6- [28,31,35,53,54] and 12-month courses [26].

**Figure 11 jcm-11-02991-f011:**
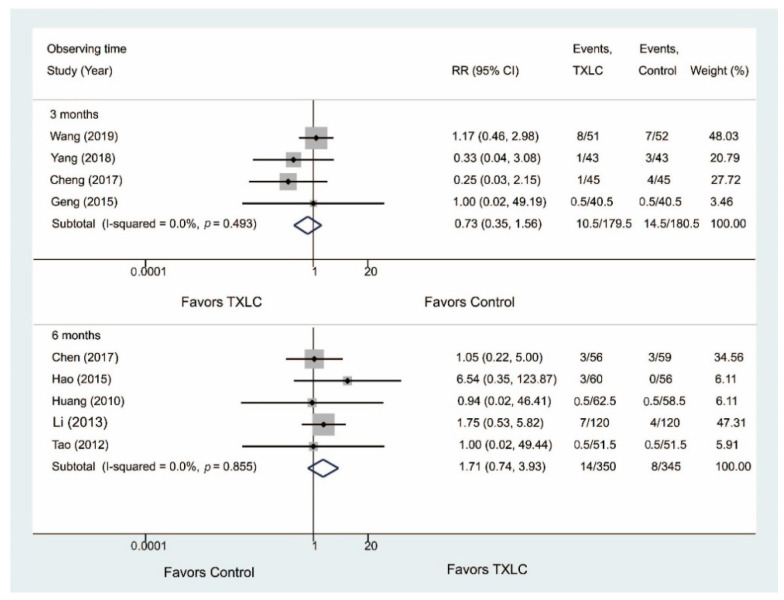
Forest plot describing the difference in adverse effects between the TXLC group and the control group on the 3- [23,27,44,48] and 6-month courses [22,30,31,33,39].

**Figure 12 jcm-11-02991-f012:**
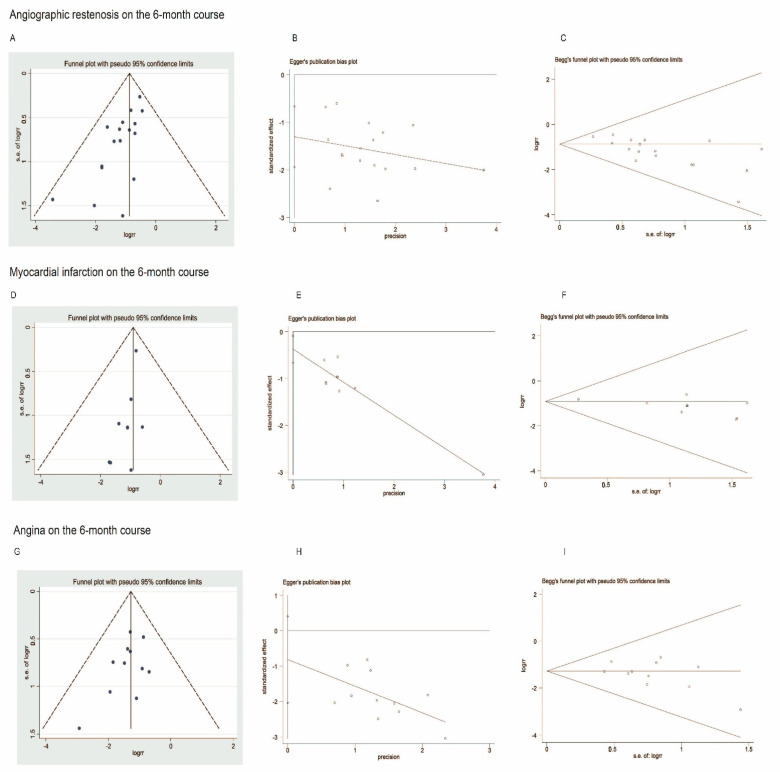
Funnel plot, Egger’s and Begg’s publication bias plots describing the occurrence of angiographic restenosis (**A**–**C**), myocardial infarction (**D**–**F**), and angina (**G**–**I**) on the 6-month course.

**Table 1 jcm-11-02991-t001:** Characteristics of the 34 trials included in the systematic review and meta-analysis.

Author (Year)	Age (Mean Age)	Gender (Male/Female)	Sample Size (TXLC/Control)	Participants	Stent Type	Treatment Courses	Observing Time	Intervention	Control	Outcomes
Chen et al., 2011 [21]	61~75 (71.1 ± 5.7)	94/66	160 (80/80)	ACS after PCI	Not reported	6 months	6 months	TXLC (1.04 g, t.i.d., p.o.) + conventional treatment	A, B, C, E, H, I	①③④⑤⑥
Chen et al., 2017 [22]	50~70 (61.90 ± 3.51)	67/48	115 (59/56)	CHD after PCI	Not reported	6 months	6 months	TXLC (0.78 g, t.i.d., p.o.) + conventional treatment	A, D, E, H, I	⑧
Cheng, 2017 [23]	55~75 (67.27 ± 5.16)	58/32	90 (45/45)	AMI after PCI	Not reported	3 months	3 months	TXLC (1.04 g, t.i.d., p.o.) + conventional treatment	A, B, C, D, E, F	⑧
Dai and Yang, 2011 [24]	45~74 (62.38 ± 6.73)	36/25	61 (31/30)	CHD after PCI	Not reported	6 months	6 months	TXLC (not reported) + conventional treatment	A, D, E	①
Deng, 2013 [25]	41~78 (55.00 ± 3.03)	41/20	61 (31/30)	CHD after PCI	Not reported	6 months	6 months	TXLC (four pills, t.i.d., p.o.) + conventional treatment	A, E, H	①
Dong and Dong, 2012 [26]	40~77 (60.13 ± 10.42)	45/18	63 (28/35)	AMI after PCI	Not reported	12 months	12 months	TXLC (four pills, t.i.d., p.o.) + conventional treatment	A, B, C, D, E, L, I	②⑤⑥⑦
Geng and Li, 2015 [27]	35~75 (54.5 ± 10.9)	49/31	80 (40/40)	ACS after PCI	Not reported	3 months	3 months	TXLC (four pills, t.i.d., p.o.) + conventional treatment	A, B, C, E, I	①⑧
Guan and Yin, 2019 [28]	36~75 (37.55 ± 3.47)	99/59	158 (79/79)	CHD after PCI	Not reported	6 months	6 months	TXLC (0.78 g, t.i.d., p.o.) + conventional treatment	A, B, C, D, E, I	①④⑤⑦
Han, 2018 [29]	45~78 (59.35 ± 6.48)	43/25	68 (34/34)	CHD after PCI	Not reported	6 months	6 months	TXLC (four pills, t.i.d., p.o.) + conventional treatment	A, E, H	①
Hao and Ning, 2015 [30]	65~81 (68.84 ± 6.87)	73/43	116 (60/56)	UA after PCI	Not reported	6 months	6 months	TXLC (three pills, t.i.d., p.o.) + conventional treatment	A, B, D, E, I	⑧
Huang, 2010 [31]	43~76 (58.3 ± 12.6)	68/52	120 (62/58)	STEMI after PCI	Not reported	6 months	6 months	TXLC (four pills, t.i.d., p.o.) + conventional treatment	A, E, H	②⑤⑥⑦⑧
Huang, 2017 [32]	25~68 (53.05 ± 5.86)	48/32	80 (40/40)	AMI after PCI	Not reported	6 months	6 months	TXLC (1.04 g, t.i.d., p.o.) + conventional treatment	A, B, C, tirofiban	②⑤
Li et al., 2013 [33]	Not report	Not reported	240 (120/120)	CHD after PCI	Not reported	6 months	6 months	TXLC (four pills, t.i.d., p.o.) + conventional treatment	A, D, E, H, tirofiban	⑧
Liang et al., 2010 [34]	40~70	52/28	80 (42/38)	AMI after PCI	Not reported	6 months	6 months	TXLC (0.52 g, t.i.d., p.o.) + conventional treatment	A, B, C, D, E, I	①②④
Liu and Jiang, 2017 [35]	45~82 (57.3 ± 19.6)	Not reported	104 (52/52)	ACS after PCI	Not reported	6 months	6 months	TXLC (three pills, t.i.d., p.o.) + conventional treatment	A, D, E	①②④⑤⑥⑦
Lu et al., 2014 [36]	45~74 (61.00 ± 7.08)	94/86	180 (90/90)	CHD after PCI	Not reported	12 months	6 months/12 months	TXLC (three pills, t.i.d., p.o.) + conventional treatment	A, D, E, placebo	①/①②④⑤
Lu, 2017 [37]	(61.8 ± 12.17)	85/45	130 (65/65)	STEMI after PCI	Not reported	6 months	6 months	TXLC (three pills, t.i.d., p.o.) + conventional treatment	Conventional treatment	①②③⑤⑥
Sun and Lang, 2018 [38]	18~85 (64.28 ± 6.85)	90/44	134 (67/67)	CHD after PCI	Not reported	6 months	6 months	TXLC (0.78 g, t.i.d., p.o.) + conventional treatment	A, E	①④⑤⑥
Tao and Zhu, 2012 [39]	(65.3 ± 12.7)	59/43	102 (51/51)	ACS after PCI	Not reported	6 months	6 months	TXLC (four pills, t.i.d., p.o.) + conventional treatment	A, B, D, E, G, I	⑧
Tian, 2017 [41]	47~79 (61.09 ± 6.30)	79/37	116 (58/58)	ACS after PCI	Not reported	6 months	6 months	TXLC (0.78 g, t.i.d., p.o.) + conventional treatment	A, D, E	①②④
Tian et al., 2014 [40]	47~75 (54.7 ± 10.02)	41/19	60 (30/30)	AMI after PCI	Not reported	3 months	6 months	TXLC (4 pills, t.i.d., p.o.) + conventional treatment	A, D, E, I	②④⑥⑦
Wang and Shi, 2009 [42]	35~74 (53.35 ± 10.49)	52/28	80 (40/40)	CHD after PCI	Not reported	6 months	6 months	TXLC (1.14 g, t.i.d., p.o.) + conventional treatment	A, E, H	①②④⑤⑥
Wang et al., 2018 [43]	39~78 (65.65 ± 9.71)	55/45	100 (50/50)	CHD after PCI	Not reported	6 months	12 months	TXLC (0.78 g, t.i.d., p.o.) + conventional treatment	A, E	①②④⑥⑧
Wang et al., 2019 [44]	40~79 (58.38 ± 16.47)	55/48	103 (51/52)	ACS after PCI	Not reported	3 months	3 months	TXLC (0.78 g, t.i.d., p.o.) + conventional treatment	A, D, E	②④⑦⑧
Wu, 2020 [45]	41~82 (59.84 ± 9.10)	35/25	60 (30/30)	CHD after PCI	Not reported	6 months	6 months	TXLC (0.78 g, t.i.d., p.o.) + conventional treatment	A, B, D, E, I	①③④
Xiao et al., 2007 [46]	(54.06 ± 10.99)	91/41	132 (62/70)	CHD after PCI	Bare metal stents	6 months	6 months	TXLC (four pills, t.i.d., p.o.) + conventional treatment	Conventional treatment	①②③④⑤⑥
Yang, 2009 [47]	40~74	47/12	59 (30/29)	STEMI after PCI	Not reported	3 months	3 months	TXLC (1.14 g, t.i.d., p.o.) + conventional treatment	A, D, E, H	②③⑤⑥⑦
Yang, 2018 [48]	56~75 (66.27 ± 6.21)	50/36	86 (43/43)	AMI after PCI	Not reported	3 months	3 months	TXLC (1 g, t.i.d., p.o.) + conventional treatment	A, B, C, D, E, F	⑧
Yao et al., 2006 [49]	35~76 (53.35 ± 12.05)	44/32	76 (38/38)	CHD after PCI	Not reported	6 months	6 months	TXLC (four pills, t.i.d., p.o.) + conventional treatment	A, E, H	①④
Zhai et al., 2019 [50]	(57.48 ± 7.95)	71/43	114 (57/57)	CHD plus T2DM after PCI	Drug-eluting stents	12 months	12 months	TXLC (four pills, t.i.d., p.o.) + conventional treatment	A, E	①②③④
Zhang and Zhang, 2009 [51]	43~70 (59.08 ± 6.61)	127/51	178 (96/82)	AMI after PCI	Not reported	24 months	3~48 months	TXLC (three pills, t.i.d., p.o.) + conventional treatment	A, B, C, D, E, I	②③④⑤⑥⑦⑧
Zhang et al., 2018 [52]	51~71 (61.85 ± 5.97)	64/56	120 (60/60)	CHD after PCI	Not reported	6 months	6 months	TXLC (three pills, t.i.d., p.o.) + conventional treatment	A, D, E	①
Zheng et al., 2010 [53]	(61.76 ± 10.62)	74/16	90 (34/56)	ACS after PCI	Not reported	6 months	6 months	TXLC (1.14 g, t.i.d., p.o.) + conventional treatment	A, B, D, E, H	②⑦
Zhou and Guo, 2007 [54]	30~76	86/50	136 (70/66)	CHD after PCI	Not reported	6 months	6 months	TXLC (three pills, t.i.d., p.o.) + conventional treatment	A, B, C, D, E, I	①②④⑦

A: aspirin; ACS: acute coronary syndrome; AMI: acute myocardial infarction; B: β-blockers; C: angiotensin-converting enzyme inhibitors; CHD: coronary heart disease; D: statins; E: clopidogrel; F: diuretic; G: calcium antagonists; g: gramme; H: low-molecular-weight heparin; I: nitrates; PCI: percutaneous coronary intervention; p.o.: per os; RCT: randomized controlled trial; STEMI: ST-segment elevation myocardial infarction; T2DM: type 2 diabetes mellitus; t.i.d.: ter in die (three times a day); TXLC: Tongxinluo capsule; UA: unstable angina; ①: occurrence of angiographic restenosis; ②: occurrence of myocardial infarction; ③: occurrence of heart failure; ④: occurrence of angina; ⑤: all-cause mortality; ⑥: mortality due to any cardiovascular event; ⑦: revascularization, including PCI and CABG; ⑧: adverse effects.

**Table 2 jcm-11-02991-t002:** The GRADE evidence profile for TXLC in the treatment of CHD patients after PCI.

Certainty Assessment							No. of Patients		Effect		Certainty	Importance
No. of Studies	Study Design	Risk of Bias	Inconsistency	Indirectness	Imprecision	Other Considerations	TXLC	Control	Relative (95% CI)	Absolute (95% CI)		
Angiographic restenosis (3 months)												
1	Randomized trial	Serious ^a^	Serious ^b^	Not serious	Serious ^c^	None	0/40 (0.0%)	2/40 (5.0%)	RR 0.20 (0.01 to 4.04)	40 fewer per 1000 (from 50 fewer to 152 fewer)	⨁◯◯◯ Very low	IMPORTANT
Angiographic restenosis (6 months)												
17	Randomized trials	Serious ^a^	Not serious	Not serious	Not serious	Publication bias strongly suspected	58/892 (6.5%)	160/880 (18.2%)	RR 0.37 (0.28 to 0.48)	115 fewer per 1000 (from 131 fewer to 95 fewer)	⨁⨁◯◯ Low	IMPORTANT
Angiographic restenosis (12 months)												
2	Randomized trials	Serious ^a^	Serious ^b^	Not serious	Not serious	None	24/147 (16.3%)	46/147 (31.3%)	RR 0.52 (0.34 to 0.80)	150 fewer per 1000 (from 207 fewer to 63 fewer)	⨁⨁◯◯ Low	IMPORTANT
Myocardial infarction (3 months)												
2	Randomized trials	Serious ^a^	Not serious	Not serious	Serious ^c^	None	0/81 (0.0%)	2/81 (2.5%)	RR 0.33 (0.04 to 3.18)	17 fewer per 1000 (from 24 fewer to 54 more)	⨁⨁◯◯ Low	IMPORTANT
Myocardial infarction (6 months)												
10	Randomized trials	Serious ^a^	Not serious	Not serious	Not serious	Publication bias strongly suspected	19/525 (3.6%)	53/543 (9.8%)	RR 0.38 (0.25 to 0.60)	61 fewer per 1000 (from 73 fewer to 39 fewer)	⨁⨁◯◯ Low	IMPORTANT
Myocardial infarction (12 months)												
3	Randomized trials	Serious ^a^	Not serious	Not serious	Serious ^c^	None	5/182 (2.7%)	11/175 (6.3%)	RR 0.44 (0.16 to 1.25)	35 fewer per 1000 (from 53 fewer to 16 more)	⨁⨁◯◯ Low	IMPORTANT
Heart failure (3 months)												
1	Randomized trial	Serious ^a^	Serious ^b^	Not serious	Serious ^c^	None	2/30 (6.7%)	8/29 (27.6%)	RR 0.24 (0.06 to 1.04)	210 fewer per 1000 (from 259 fewer to 11 more)	⨁◯◯◯ Very low	IMPORTANT
Heart failure (6 months)												
4	Randomized trials	Serious ^a^	Not serious	Not serious	Not serious	None	13/237 (5.5%)	42/245 (17.1%)	RR 0.32 (0.18 to 0.56)	117 fewer per 1000 (from 141 fewer to 75 fewer)	⨁⨁⨁◯ Moderate	IMPORTANT
Heart failure (12 months)												
1	Randomized trial	Serious ^a^	Serious ^b^	Not serious	Serious ^c^	None	1/57 (1.8%)	2/57 (3.5%)	RR 0.50 (0.05 to 5.36)	18 fewer per 1000 (from 33 fewer to 153 more)	⨁◯◯◯ Very low	IMPORTANT
Angina (3 months)												
1	Randomized trial	Serious ^a^	Serious ^b^	Not serious	Serious ^c^	None	1/51 (2.0%)	4/52 (7.7%)	RR 0.25 (0.03 to 2.20)	58 fewer per 1000 (from 75 fewer to 92 more)	⨁◯◯◯ Very low	IMPORTANT
Angina (6 months)												
11	Randomized trials	Serious ^a^	Not serious	Not serious	Not serious	None	27/618 (4.4%)	106/618 (17.2%)	RR 0.26 (0.17 to 0.38)	127 fewer per 1000 (from 142 fewer to 106 fewer)	⨁⨁⨁◯ Moderate	IMPORTANT
Angina (12 months)												
2	Randomized trials	Serious ^a^	Not serious	Not serious	Not serious	None	10/147 (6.8%)	25/147 (17.0%)	RR 0.40 (0.20 to 0.80)	102 fewer per 1000 (from 136 fewer to 34 fewer)	⨁⨁⨁◯ Moderate	IMPORTANT
All-cause mortality (3 months)												
1	Randomized trial	Serious ^a^	Serious ^b^	Not serious	Serious ^c^	None	0/30 (0.0%)	1/29 (3.4%)	RR 0.32 (0.01 to 7.61)	23 fewer per 1000 (from 34 fewer to 228 more)	⨁◯◯◯ Very low	IMPORTANT
All-cause mortality (6 months)												
9	Randomized trials	Serious ^a^	Not serious	Not serious	Not serious	None	4/548 (0.7%)	23/552 (4.2%)	RR 0.24 (0.10 to 0.58)	32 fewer per 1000 (from 38 fewer to 18 fewer)	⨁⨁⨁◯ Moderate	IMPORTANT
All-cause mortality (12 months)												
2	Randomized trials	Serious ^a^	Not serious	Not serious	Serious ^c^	None	1/125 (0.8%)	2/118 (1.7%)	RR 0.51 (0.07 to 3.67)	8 fewer per 1000 (from 16 fewer to 45 more)	⨁⨁◯◯ Low	IMPORTANT
Mortality due to any cardiovascular event (3 months)												
1	Randomized trial	Serious ^a^	Serious ^b^	Not serious	Serious ^c^	None	0/30 (0.0%)	1/29 (3.4%)	RR 0.32 (0.01 to 7.61)	23 fewer per 1000 (from 34 fewer to 228 more)	⨁◯◯◯ Very low	IMPORTANT
Mortality due to any cardiovascular event (6 months)												
7	Randomized trials	Serious ^a^	Not serious	Not serious	Not serious	None	2/429 (0.5%)	13/433 (3.0%)	RR 0.27 (0.09 to 0.80)	22 fewer per 1000 (from 27 fewer to 6 fewer)	⨁⨁⨁◯ Moderate	IMPORTANT
Mortality due to any cardiovascular event (12 months)												
1	Randomized trial	Serious ^a^	Serious ^b^	Not serious	Serious ^c^	None	1/35 (2.9%)	2/28 (7.1%)	RR 0.40 (0.04 to 4.19)	43 fewer per 1000 (from 69 fewer to 228 more)	⨁◯◯◯ Very low	IMPORTANT
Revascularization (3 months)												
2	Randomized trials	Serious ^a^	Not serious	Not serious	Serious ^c^	None	1/81 (1.2%)	2/81 (2.5%)	RR 0.60 (0.08 to 4.47)	10 fewer per 1000 (from 23 fewer to 86 more)	⨁⨁◯◯ Low	IMPORTANT
Revascularization (6 months)												
5	Randomized trials	Serious ^a^	Not serious	Not serious	Not serious	None	6/297 (2.0%)	34/311 (10.9%)	RR 0.20 (0.09 to 0.46)	87 fewer per 1000 (from 99 fewer to 59 fewer)	⨁⨁⨁◯ Moderate	IMPORTANT
Revascularization (12 months)												
1	Randomized trials	Serious ^a^	Serious ^b^	Not serious	Serious ^c^	None	0/35 (0.0%)	3/28 (10.7%)	RR 0.12 (0.01 to 2.14)	94 fewer per 1000 (from 106 fewer to 122 more)	⨁◯◯◯ Very low	IMPORTANT
Adverse effects (3 months)												
4	Randomized trials	Serious ^a^	Not serious	Not serious	Serious ^c^	None	10/179 (5.6%)	14/180 (7.8%)	RR 0.73 (0.35 to 1.56)	21 fewer per 1000 (from 51 fewer to 44 more)	⨁⨁◯◯ Low	IMPORTANT
Adverse effects (6 months)												
5	Randomized trials	Serious ^a^	Not serious	Not serious	Serious ^c^	None	14/350 (4.0%)	8/345 (2.3%)	RR 1.71 (0.74 to 3.93)	16 more per 1000 (from 6 fewer to 68 more)	⨁⨁◯◯ Low	IMPORTANT

CI, confidence interval; RR, risk ratio; ^a^ All of the studies mentioned random methods. Sixteen studies reported the specific random method, one study reported the allocation concealment method, 33 studies reported the attrition bias method. No study reported the blinding of participants and personnel or outcome assessment method. ^b^ Moderate heterogeneity was detected or not estimable. ^c^ No statistical significance or small sample size that does not meet the optimal information size. ⨁◯◯◯: the certainty of the evidence was very low; ⨁⨁◯◯: the certainty of the evidence was low; ⨁⨁⨁◯: the certainty of the evidence was moderate.

## Data Availability

The datasets used and/or analyzed in the current study are available from the corresponding author upon reasonable request.

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
