# Peer review of "Efficacy and Safety of Different Courses of Tongxinluo Capsule as Adjuvant Therapy for Coronary Heart Disease after Percutaneous Coronary Intervention: A Systematic Review and Meta-Analysis of Randomized Controlled Trials"

_jcm, 2022, doi:10.3390/jcm11112991_

Round 1

Reviewer 1 Report

Drs Jiaqi Hui and colleagues present a meta-analysis to evaluate the efficacy and safety of different courses of TXLC for CHD after percutaneous coronary intervention.

Overall, this meta-analysis is well structured and coherent, taking account that TXLC is only used in traditional Chinese medicine. However, how could be useful this meta-analysis for international community?

In fact, the disease could be difference between different races or regions. Please add in discussion some points about races and regions differences and its possible beneficial effects of TXLC. 

Author Response

Response 1: We do appreciate the reviewer’s comment. Based on the reality, most of the research on Chinese medicine is conducted in China, resulting in the inevitably limitations of conclusions. In this regard, we have discussed objectively both in the Discussion (Line 422-427) and the Limitations (Line 473-475, 481) in the updated submitted manuscript. Besides, we have conducted comprehensive searches in the designated database without restricting language or ethnicity (Line 105). Since proprietary Chinese medicines have not been promoted globally, all participants ultimately included were Chinese due to the limitation of application scope. As for the impact, Chinese medicine has been widely used from the very ancient time in many countries outside China, such as Korea, Japan and Southeast Asian countries. Furthermore, in a globalized world, effective medicine including Chinese medicine no doubt will gain popularity and global impact. We don’t doubt that studies like ours, along with contributions from you and other researchers, will play a positive role in promoting the globalization of effective medicines for human health.

Reviewer 2 Report

This study aims to address the effectiveness of the Tongxinluo capsule (TXLC) for cardiovascular heart disease (CHD) after percutaneous coronary intervention (PCI). 

The meta-analysis was performed using a fixed- or random-effects model. The risk of adverse cardiovascular events, mortality or adverse effects was assessed by hazard ratio (RR) with 95% confidence interval (CI). In the end, 34 studies involving 3652 patients were included. After a treatment of 6 months, TXLC combined with conventional treatment alone achieved better efficacy in reducing the risk of angiographic restenosis (RR = 0.37, 95% CI [0.28, 0.48], p < 0.001), myocardial infarction (RR = 0.38, 95% CI [0.25, 0.60], p < 0.001), heart failure (RR = 0. 32, 95% CI [0.18, 0.56], p < 0.001), angina (RR = 0.26, 95% CI [0.17, 0.38], p < 0.001), revascularisation (RR = 0.20, 95% CI [0.09, 0.46], p < 0.001), all-cause mortality. 001), all-cause mortality (RR = 0.24, 95% CI [0.10, 0.58], p = 0.001) and mortality due to any cardiovascular event (RR = 0.27, 95% CI [0.09, 0.80], p = 0.018). After the 12-month treatment, TXLC reduced the risk of angina recurrence (RR = 0.40, 95% CI [0.20, 0.80], p = 0.009). However, no difference was found in terms of outcomes after treatment at 3 months. Furthermore, no difference was found in the incidence of adverse effects after treatment at 3 and 6 months (3 months: RR = 0.73, 95% CI [0.35, 1.56], p = 0.418; 6 months: RR = 1.71, 95% CI [0.74, 3.93], p = 0.209).

As specified in the study, the risk of BIAS is very high mainly due to the failure to classify patients according to age, sex, and race despite meeting perfect enrolment criteria.

Therefore, we agree with the limitations in suggesting that future studies on the adjuvant treatment of CHD after PCI with TXLC should pay more attention to random sequence generation, allocation concealment, and implementation of blinding. More large-scale, prospective, randomized, double-blind, and long-term RCTs need to be conducted to improve the methodological flaws of the included studies. Furthermore, different stent types, treatment duration, and dose-effect relationships should be considered non-negligible factors influencing the efficacy and safety of TXLC. It is suggested that future RCTs pay attention to the incidence of adverse cardiovascular events, mortality, and adverse reactions in patients with long-term follow-up to provide better data to guide clinical decision-making.

However, I consider the article to be very original and well-structured, clear, and well organized in terms of vocabulary and choice of scientific English.

Author Response

Response 1: We do appreciate the reviewer’s kind suggestion. We have added relevant references to describe the influence of different races or genders on the cardiovascular events in patients with coronary heart disease after PCI (Line 422-427). However, in total, the included 3652 patients in this review were all from China, of which 2072 were males and 1236 were females, and two studies did not clarify the number of males or females. The age of patients ranged from 18 to 85 years old. It was not possible to classify patients according to age, sex, and race, and subgroup was also impossible to conduct because of insufficient numbers and the not high quality of included trials. Therefore, we have suggested that more large-scale, prospective, randomized, double-blind, and long-term RCTs are needed to be conducted to improve the methodological flaws of the included studies. Besides, different stent types, the treatment duration, and the dose-effect relationship, and racial differences should be considered as non-negligible factors affecting the effi-cacy and safety of TXLC. Future RCTs are suggested to pay attention to the incidence of adverse cardiovascular events, mortality, and adverse reactions in patients with long-term follow-up to pro-vide better data to guide clinical decision-making.